

# Dynamic residue interaction network analysis of the oseltamivir binding site of N1 neuraminidase and its H274Y mutation site conferring drug resistance in influenza A virus

Mohini Yadav[1], Manabu Igarashi[2,3] and Norifumi Yamamoto[1]

[1] Department of Applied Chemistry, Faculty of Engineering, Chiba Institute of Technology, Narashino, Japan
[2] Division of Global Epidemiology, Research Center for Zoonosis Control, Hokkaido University, Sapporo, Japan
[3] International Collaboration Unit, Research Center for Zoonosis Control, Hokkaido University, Sapporo, Japan

## ABSTRACT

**Background**. Oseltamivir (OTV)-resistant influenza virus exhibits His-to-Tyr mutation at residue 274 (H274Y) in N1 neuraminidase (NA). However, the molecular mechanisms by which the H274Y mutation in NA reduces its binding affinity to OTV have not been fully elucidated.

**Methods**. In this study, we used dynamic residue interaction network (dRIN) analysis based on molecular dynamics simulation to investigate the correlation between the OTV binding site of NA and its H274Y mutation site.

**Results**. dRIN analysis revealed that the OTV binding site and H274Y mutation site of NA interact via the three interface residues connecting them. H274Y mutation significantly enhanced the interaction between residue 274 and the three interface residues in NA, thereby significantly decreasing the interaction between OTV and its surrounding loop 150 residues. Thus, we concluded that such changes in residue interactions could reduce the binding affinity of OTV to NA, resulting in drug resistant influenza viruses. Using dRIN analysis, we succeeded in understanding the characteristic changes in residue interactions due to H274Y mutation, which can elucidate the molecular mechanism of reduction in OTV binding affinity to influenza NA. Finally, the dRIN analysis used in this study can be widely applied to various systems such as individual proteins, protein-ligand complexes, and protein-protein complexes, to characterize the dynamic aspects of the interactions.

## INTRODUCTION

Influenza is a highly contagious respiratory disease caused by influenza virus that can result in mild to severe illness and even death (*Palese, 2004*). Hemagglutinin (HA) and neuraminidase (NA) are glycoproteins that play essential roles in the replication of influenza virus (*Gamblin & Skehel, 2010*). HA is a receptor-binding glycoprotein that binds to the

Corresponding author
Norifumi Yamamoto,
norifumi.yamamoto@it-chiba.ac.jp

sialic acid at the end of a sugar chain present on the surface of host cells, thus facilitating viral entry into cells via endocytosis. NA functions as a scissor to cleave the bond between sialic acid and cellular glycoconjugates in the final stages of infection, resulting in the release of progeny viruses that then infect the surrounding cells. Both HA and NA can be targeted by inhibitor molecules that may be developed as anti-influenza drugs. Indeed, anti-NA drugs that inhibit the interaction between NA and sialic acid have been developed (*Moscona, 2005*).

Oseltamivir (OTV) is a leading NA inhibitor that is widely used in the treatment and prevention of influenza (*Kim et al., 1997*). However, the emergence of virus strains resistant to OTV has raised many concerns (*De Jong et al., 2005*). For example, in the 2007–2008 influenza season, OTV-resistant strains emerged worldwide, and within a year, these resistant strains were present in most cases of seasonal H1N1 influenza infection (*Moscona, 2009*). These OTV-resistant strains are characterized by a His-to-Tyr mutation at NA residue 274 (H274Y) (*Gubareva et al., 2001*; *Wang, Tai & Mendel, 2002*; *Ives et al., 2002*; *Hurt et al., 2009*), located near but not in the OTV binding site of NA (*Russell et al., 2006*). The crystal structure of the complex formed by OTV and H274Y mutant NA exhibits a subtle conformational change, compared to that formed by OTV and WT NA, wherein the carboxylate side chain of the E276 adjacent to residue 274 is shifted to the OTV binding site (*Collins et al., 2008*). As E276 is located at the OTV binding site, a conformational change due to H274Y mutation disrupts favorable interactions between OTV and the binding site residues in NA, resulting in reduced binding affinity of OTV to NA and thus conferring drug resistance to influenza virus.

Molecular dynamics (MD) simulations have provided theoretical perspectives on OTV resistance in influenza virus (*Malaisree et al., 2009*; *Wang & Zheng, 2009*; *Park & Jo, 2009*; *Nguyen, Mai & Li, 2011*; *Li et al., 2012*; *Vergara-Jaque et al., 2012*; *Woods et al., 2012*; *Ripoll et al., 2012*; *Woods et al., 2013*; *Yusuf et al., 2016*). For example, *Malaisree et al. (2009)* used MD simulations to show that the H274Y mutation in NA causes the bulkier side chain of tyrosine to rotate the carboxylate group in E276 by 115°, thereby reducing hydrophobicity and pocket size; because of changes in the binding site, the pentyloxy side chain of OTV rotates by 125°, decreasing the binding free energy by approximately 5 kcal mol$^{-1}$ (*Malaisree et al., 2009*). Thus, previous studies have investigated conformational changes at the binding site in NA using MD simulations. However, studies focusing only on the endpoint of the ligand-protein interaction based on three-dimensional structures may not be sufficient to explain the impact of mutations on drug resistance. It is unclear whether the decrease in the binding affinity of OTV to NA is influenced by a correlation between the binding site and its H274Y mutation site.

Residue interaction networks (RINs) have previously been used to describe protein spatial structures as networks of interactions among amino acid residues (*Paola et al., 2013*; *Csermely et al., 2013*; *Yan et al., 2014*; *Shcherbinin & Veselovsky, 2019*). In a protein, there are covalent and non-covalent interactions among amino acid residues that stabilize its three-dimensional structure and determine the functions required in living organisms. RIN visualizes interactions within the spatial architecture of a protein as a simple graph with nodes and edges representing the residue and inter-residue interactions, respectively.

Many types of RIN have been proposed, depending on the interactions among residues (*Paola et al., 2013*; *Csermely et al., 2013*; *Yan et al., 2014*; *Shcherbinin & Veselovsky, 2019*). One widely used approach of RIN is the identification of different types of physicochemical interactions between residues according to certain criteria (*Piovesan, Minervini & Tosatto, 2016*; *Shcherbinin & Veselovsky, 2019*). Currently, most RIN approaches are static, built on a single protein structure and disregarding the dynamic properties of proteins, including the creation and annihilation of non-covalent interactions. Several studies have used dynamic RIN (dRIN), which is constructed using a set of multiple protein structures, obtained through MD simulations, to reveal the ensemble-averaged and dynamic features of residue interactions (*Pasi et al., 2012*; *Bhattacharyya, Bhat & Vishveshwara, 2013*; *Tiberti et al., 2014*; *Brown et al., 2017*; *Serçinoğlu & Ozbek, 2018*; *Contreras-Riquelme et al., 2018*). The original static RIN and its extended version, dRIN, have been used in various analyses such as protein stability (*Brinda & Vishveshwara, 2005*; *Giollo et al., 2014*), allosteric behavior (*Sethi et al., 2009*) and drug resistance (*Xue et al., 2012*; *Bhakat, Martin & Soliman, 2014*; *Xue et al., 2014*; *Zhang et al., 2019*). Recently, *Buthelezi et al. (2019)* used RIN analysis to obtain a molecular perspective of drug resistance in influenza viruses with H274Y mutation in NA; however, they only examined the changes in residue interactions with the mutation based on static RINs constructed from representative average structures obtained via MD simulations (*Buthelezi et al., 2019*). Overall, the mechanism by which drug resistance conferring H274Y mutation in NA of influenza virus affects the dynamic behavior of residue interactions remains unclear.

In this study, we investigated the changes in residue-residue and residue-ligand interactions associated with the H274Y mutation in the complex of OTV bound to influenza virus NA using dRIN analysis. The method used in this study extends the original static version to construct dRINs from multiple protein structures based on MD trajectories, providing a statistical perspective on residue interactions. Here, we successfully clarified the correlation between the OTV binding site of N1 NA and its H274Y mutation site that confers drug resistance in influenza virus. This study provides novel theoretical insights into the molecular mechanism underlying OTV resistance in influenza virus caused by H274Y mutation in NA.

## METHODS

### Initial structures

Structure preparation and MD simulations were performed using the Amber 20 package (*Case et al., 2020*). The crystal structures of wild-type (WT) avian influenza virus A/H5N1 NA and its H274Y mutant in complex with OTV were obtained from the Protein Data Bank (PDB code: 2HU4 and 3CL0) (*Russell et al., 2006*; *Collins et al., 2008*), using a single monomer for modeling. H5N1 NA contains one calcium ion that is necessary for structural stability (*Smith et al., 2006*). As no calcium ions were found in the crystal structure of H5N1 NA registered as 2HU4, the coordinates of the calcium ion were obtained from the corresponding structure registered as 3CL0. The protonation state of His in the modeled complex at pH 7 was determined using the PDB2PQR server (*Dolinsky et al., 2004*), and
the other ionized residues, that is, Arg, Lys, Asp, and Glu, were treated as charged entities. The LEaP module in Amber 20 was used to add missing hydrogen atoms to the proteins and OTV. H5N1 NA contains eight disulfide bonds. For each disulfide bond, the "bond" command was executed in the tLEaP program to create a covalent bond between the SG atoms of the proximate cysteine residues. The FF14SB variant of the AMBER force field was used to describe the proteins (*Maier et al., 2015*). The generalized AMBER force field was applied to OTV (*Wang et al., 2004*). The partial atomic charges in OTV were determined according to the restrained electrostatic potential fitting procedure (*Bayly et al., 1993*), based on the quantum chemistry calculations at the HF/6-31G(d) level with the Gaussian 16 program (*Frisch et al., 2016*). The complexes of the WT and H274Y mutant NA with OTV were solvated in a truncated octahedral box of TIP3P water molecules with a distance of at least 10 Å around them. The total charges of the complexes were neutralized by adding sodium counter ions. The total number of atoms in the complexes of the WT and H274Y mutant NA were 30,252 and 29,635, respectively.

## Molecular dynamics (MD) simulations

Each system was energy minimized using the steepest descent method for 500 steps, followed by the conjugate gradient method for 4,500 steps, with a harmonic restraint weight of 10 kcal mol$^{-1}$ Å$^{-2}$ on the complexes, except for hydrogen atoms. After energy minimization, each system was gradually heated to 300 K over a period of 200 ps in the *NVT* ensemble. All MD simulations were performed using the PMEMD module in the AMBER 20 package. Temperature was regulated using the weak-coupling algorithm (*Berendsen, Postma & Funsteren, 1984*). All bond lengths including hydrogen atoms were constrained using the SHAKE algorithm (*Ryckaert, Ciccotti & Berendsen, 1977*), allowing an MD time-step of 2 fs to be used. Periodic boundary conditions were adopted. A cutoff of 8 Å was set for non-bonded interactions. Long-range electrostatics were treated using the particle-mesh Ewald method (*Darden, York & Pedersen, 1993*). After heating, MD simulations were performed for 80 ns in the *NpT* ensemble at a pressure of 1.0 atm and a temperature of 300 K. The pressure was maintained using a Berendsen barostat. The production phase to be analyzed was the last 40 ns of MD simulations, which was determined based on the root mean square displacement (RMSD) for the backbone atoms in the proteins with respect to the initial structure along the simulation time. The time series of RMSD for the backbone atoms in the WT and H274Y mutant NA are shown in Fig. 1. The changes in RMSD were almost constant after 40 ns, indicating that the MD simulations properly converged in this time region.

## Binding free energy calculations

Binding free energies were estimated for OTV complexed with WT and H274Y mutant NA using the Molecular Mechanics Poisson Boltzmann Surface Area (MM-PBSA) method in the Amber 20 package, MMPBSA.py (*Miller et al., 2012*). The electrostatic contribution to the solvation free energy was estimated using the adaptive Poisson Boltzmann (PB) solver (*Baker et al., 2001*). The dielectric constants in the protein and water were set to 4 and 80, respectively. A relatively large dielectric constant is desirable in NA, considering that its
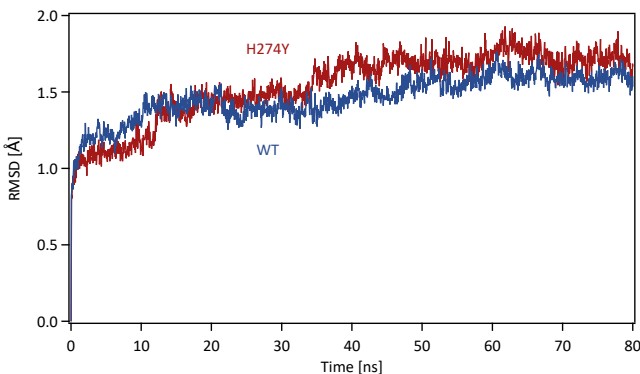

**Figure 1** **Time series of the root mean square displacement (RMSD).** RMSDs for the backbone atoms in the WT and H274Y mutant NA with respect to the initial structure along the simulation time. The production phase to be analyzed was the last 40 ns of MD simulations.

binding site contains many charged residues (*Hou et al., 2011*). The ionic strength was set at 150 mM. The ratio between the longest dimension of the rectangular finite-difference grid and that of the solute was set to 4. The linear PB equation was solved using a maximum of 1000 iterations. MM-PBSA calculations were performed over 2,000 frames extracted from the production phase in the last 40 ns of MD simulations.

The entropy due to the vibrational degrees of freedom was calculated by normal mode analysis of 100 configurations using the mmpbsa_py_nabnmode program in the Amber 20 package. Each configuration was energy minimized with a generalized Born solvent model, using a maximum of 10,000 steps with a target root-mean-square gradient of $10^{-3}$ kcal $mol^{-1}$ $Å^{-1}$.

## Dynamic residue interaction network (dRIN) analysis

Residue interactions were analyzed for 2,000 three-dimensional structures of the WT and H274Y mutant NA-OTV complexes generated in the 40-ns production phase of the MD simulations. In this study, ligand-residue interactions were included in the residue interactions. The RING software was used to categorize residue-residue and residue-ligand interactions into specific types, such as hydrogen bonds, van der Waals (vdW) interactions, disulfide bonds, salt bridges, $\pi$-cation interactions, and $\pi-\pi$ stacking interactions (*Piovesan, Minervini & Tosatto, 2016*). As the RING software missed hydrogen bonds between residues and the ligand in the NA-OTV complexes, the cpptraj program in the Amber 20 package was used to detect hydrogen bonds. Although the RING software identified several different types of interactions per residue pair, only one interaction per interaction type was considered. We further investigated the interactions between residues that were not included in the attractive interactions identified by the RING software but were in close contact with each other. In this study, two residues were identified as being in close contact, if $d_{ij} - (\sigma_i + \sigma_j) < 0.4$ Å, where $d_{ij}$ denotes the interatomic distance between the $i$- and $j$-th atoms in the two residues and $\sigma_i$ is the vdW radius of the $i$-th atom. This definition of close contact has been adopted by several programs for the visualization

and analysis of molecular structures, such as the UCSF Chimera software (*Pettersen et al., 2004*). By repeatedly applying the above procedure, the dRINs were constructed based on the sets of residue interactions and close contacts obtained from the MD simulations for the WT and H274Y mutant NA-OTV complexes. The occupancy of residue interactions was calculated as the percentage of frames with interactions between residues in the MD simulation. The dRINs were visualized using Cytoscape software (*Shannon et al., 2003*) with residues and their interactions represented by nodes and edges, respectively.

## RESULTS

### Binding structures and energies

Figure 2 represents snapshot images obtained from the MD simulations for the WT and H274Y mutant NA-OTV complexes, showing the OTV binding site and the region adjacent to residue 274. Table 1 summarizes the computed binding free energies ($\Delta G$) of OTV for the WT and H274Y mutant NA obtained from the MM-PBSA calculations, along with the enthalpy ($\Delta H$) and entropy ($T\Delta S$). The detailed results for the energetic components divided by each intermolecular interaction are shown in Table S1. The binding free energies of OTV were computed to be $-11.54$ and $-4.34$ kcal mol$^{-1}$ for the WT and H274Y mutant NA, respectively. The 7.20 kcal mol$^{-1}$ increase in the binding free energy of OTV due to the H274Y mutation could significantly reduce the efficiency of this inhibitor against NA. This is supported by the experimental fact that the H274Y mutant NA has a 300- to 1700-fold decrease in relative susceptibility to OTV compared with the WT NA in H5N1 viruses (*Ives et al., 2002*; *Wang, Tai & Mendel, 2002*; *Kiso et al., 2004*; *De Jong et al., 2005*; *Mishin, Hayden & Gubareva, 2005*; *Yen et al., 2007*). Several computational studies have been conducted on the change in the binding free energy of OTV due to the H274Y mutation (*Malaisree et al., 2009*; *Wang & Zheng, 2009*; *Park & Jo, 2009*; *Nguyen, Mai & Li, 2011*; *Li et al., 2012*; *Vergara-Jaque et al., 2012*; *Woods et al., 2012*; *Ripoll et al., 2012*; *Woods et al., 2013*; *Yusuf et al., 2016*). The current results are qualitatively consistent with previous experimental and computational studies, indicating that the MD simulations, which form the basis for the subsequent analyses, were reliable.

The binding free energy computed using the MM-PBSA approach can be sensitive to numerous factors, such as the value of the dielectric constant ($\varepsilon$) chosen for the protein. Generally, $\varepsilon = 1.0$ is used for proteins; however, in this study, $\varepsilon = 4.0$ was used for NA based on previous studies, considering that the binding site of NA contains many charged residues (*Hou et al., 2011*). The results of binding free energies obtained using the MM-PBSA method with $\varepsilon = 1.0$ to confirm the validity of this factor are shown in Table S2. The results calculated with $\varepsilon = 1.0$ showed a larger difference in binding free energies (10.48 kcal mol$^{-1}$) than those calculated with $\varepsilon = 4.0$ (7.20 kcal mol$^{-1}$), further reducing the affinity of NA for OTV after the H274Y mutation. The difference in the binding free energy of the WT and H274Y mutant NA to OTV was calculated to be approximately 4.4 kcal mol$^{-1}$, based on the experimentally measured IC$_{50}$ value (*Yen et al., 2007*). This clearly indicates that it is better to use $\varepsilon = 4.0$ for NA in the MM-PBSA calculation, as mentioned in the previous study (*Hou et al., 2011*).

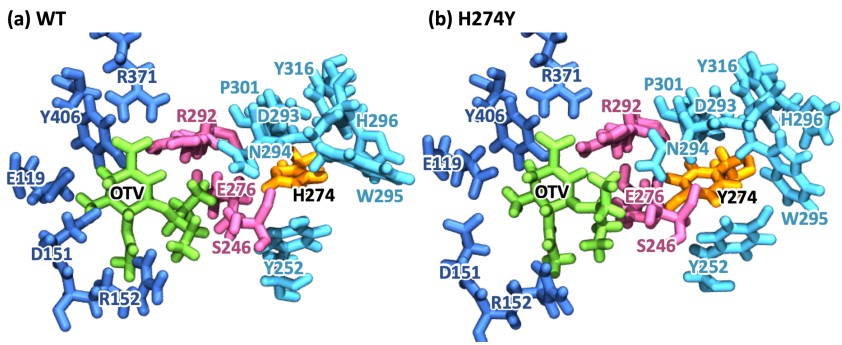

**(a) WT**

**(b) H274Y**

**Figure 2  Snapshot images of the oseltamivir-neuraminidase complexes.** Snapshot images obtained from the MD simulations for (A) the wild-type (WT) and (B) H274Y mutant influenza neuraminidase, showing the oseltamivir (OTV) binding site and the mutation site. OTV is represented in green, and the residue 274 is indicated in orange. The residues around OTV and the residue 274 can be divided into three groups. Two of the three are groups of residues that interact directly with OTV or the residue 274 alone, which are represented in blue and cyan, respectively. The other is a group of residues that interact with both OTV and residue 274, which is represented in purple.

**Table 1  Calculated binding free energies for oseltamivir to the wild-type (WT) and H274Y mutant influenza neuraminidase obtained from the MM-PBSA calculations.**

|  | $\Delta H$ (kcal mol$^{-1}$) | $T\Delta S$ (kcal mol$^{-1}$) | $\Delta G$ (kcal mol$^{-1}$) | $\Delta\Delta G$ (kcal mol$^{-1}$) |
|---|---|---|---|---|
| WT | $-35.19 \pm 0.07$ | $-23.65 \pm 0.47$ | $-11.54 \pm 0.48$ | |
| H274Y | $-27.28 \pm 0.09$ | $-22.94 \pm 0.50$ | $-4.34 \pm 0.51$ | 7.20 |

To further analyze the structural changes associated with the H274Y mutation, the pocket cavity volumes of the WT and mutant NA were examined using the POVME 3.0 software (*Wagner et al., 2017*). The results analyzed for the last 40 ns of the MD simulations showed that the average pocket cavity volumes were computed to be 394 and 853 Å$^3$, for the WT and H274Y mutant NA, respectively. The pocket cavity volume of the H274Y mutant NA was increased more than twofold compared to that of the WT NA. In the following sections, such structural changes in the drug-binding site of NA have been analyzed in more detail in terms of residue interactions.

### Dynamic residue interaction network (dRIN) analysis

Figure 3 shows the dRIN graphs for the WT and H274Y mutant NA complexed with OTV. In a dRIN graph, a node represents an amino acid residue or a ligand, and an edge connecting two nodes represents a residue-residue or a residue-ligand interaction. In most current approaches, RIN is modeled using a single structure (*Piovesan, Minervini & Tosatto, 2016*; *Shcherbinin & Veselovsky, 2019*). dRIN extends the original version as it models using multiple structures, providing a statistical perspective on residue interactions. The thickness of the edge represents the occupancy at which an interaction occurs between residues. In Fig. 3, only residue interactions with total occupancy > 10% are shown.

Figure 4A shows the occupancies at which some interactions occur between residues. Occupancies for each type of residue interaction are summarized in Tables 2 and 3. Figure
_________________________________________________

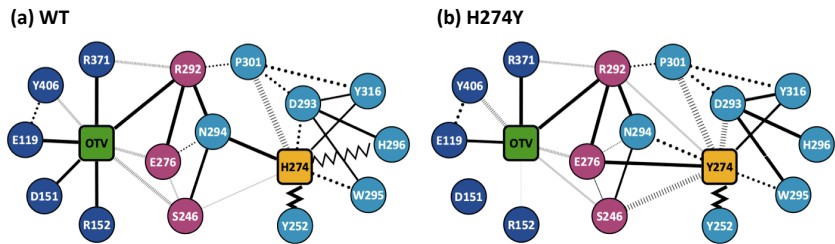

**Figure 3 Dynamic residue interaction network (dRIN) graphs.** dRINs for (A) the wild-type (WT) and (B) H274Y mutant influenza neuraminidase, showing the oseltamivir (OTV) binding site and mutation site. The node corresponding to OTV is represented in green, and the node corresponding to residue 274 is shown in orange. The type of each edge represents the type of residue interactions. A solid edge represents a hydrogen bond, a dotted edge indicates a vdW interaction, a zigzag indicates $\pi-\pi$ stacking interaction, and a dashed edge indicates a close contact. The thickness of the edge represents the occupancy at which an interaction occurs between residues. The residues around OTV and residue 274 can be divided into three groups. Two of the three are groups of residues that interact directly with OTV or the residue 274 alone, which are represented in blue and cyan, respectively. The other is a group of residues that interact with both OTV and residue 274, which is represented in purple.

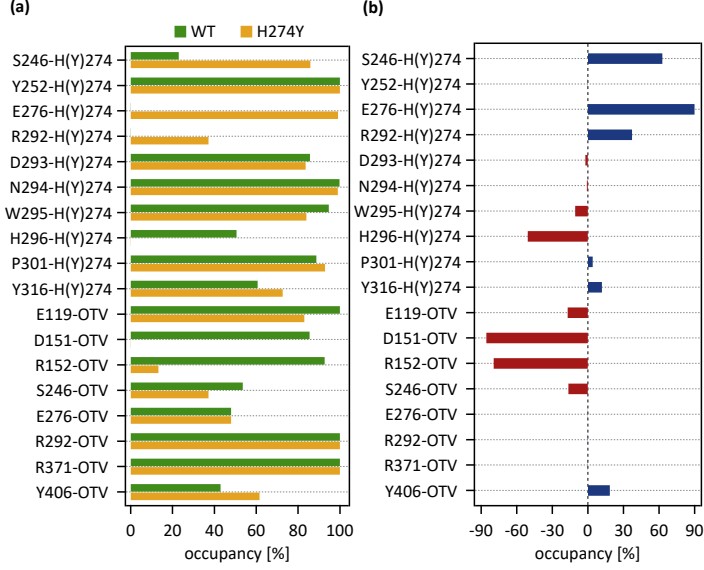

**Figure 4 Occupancies at which some interactions are formed between residues.** (A) Occupancies of residue interactions for the wild-type (WT) and H274Y mutant within the oseltamivir (OTV) binding site and mutation site. (B) Changes in the occupancy of residue interactions due to H274Y mutation.

4B shows the change in the occupancies of residues interacting with each of the WT and H274Y mutant NA.

Figure 5A shows the decomposed protein-ligand complex binding free energies of the WT and H274Y mutant NA on a per-residue basis according to the MM-PBSA calculations. Detailed information on the analysis of the protein-ligand interaction, divided

**Table 2 Occupancies at which some interactions are formed between residues in the complex of wild-type influenza neuraminidase and oseltamivir (OTV).** Residue interaction is classified as hydrogen bond (HB), van der Waals interaction (vdW), $\pi - \pi$ stacking interaction ($\pi\pi$), and close contact (CC).

| Residue pair | HB [%] | vdW [%] | $\pi\pi$ [%] | CC [%] | Total [%] |
|---|---|---|---|---|---|
| S246-H274 | 5 | 8 | 0 | 12 | 23 |
| Y252-H274 | 6 | 55 | 100 | 0 | 100 |
| E276-H274 | 0 | 0 | 0 | 0 | 0 |
| R292-H274 | 0 | 0 | 0 | 0 | 0 |
| D293-H274 | 0 | 48 | 0 | 37 | 86 |
| N294-H274 | 98 | 85 | 0 | 1 | 100 |
| W295-H274 | 12 | 93 | 53 | 1 | 95 |
| H296-H274 | 1 | 42 | 49 | 1 | 51 |
| P301-H274 | 0 | 17 | 0 | 72 | 89 |
| Y316-H274 | 55 | 0 | 0 | 6 | 61 |
| E119-OTV | 100 | 13 | 0 | 0 | 100 |
| D151-OTV | 85 | 15 | 0 | 1 | 86 |
| R152-OTV | 85 | 0 | 0 | 8 | 93 |
| S246-OTV | 0 | 0 | 0 | 54 | 54 |
| E276-OTV | 0 | 0 | 0 | 48 | 48 |
| R292-OTV | 100 | 0 | 0 | 0 | 100 |
| R371-OTV | 100 | 0 | 0 | 0 | 100 |
| Y406-OTV | 0 | 9 | 0 | 35 | 43 |

by component per residue, is summarized in Tables 4 and 5. Figure 5B shows the change in the decomposed binding free energies between the WT and H274Y mutant NA.

### dRIN of WT NA

Figure 3A shows that the OTV binding site in WT NA has five charged residues, E119, D151, R152, R292, and R371, which form strong hydrogen-bond interactions with OTV. Of these, the hydrogen-bond interaction with three charged residues, E119, R292, and R371 were very robust, accounting for 100% of the interactions formed; the hydrogen-bond interaction with the remaining two of the five charged residues, D151 and R152, have slightly lower occupancies, around 90% (Fig. 4A and Table 2). As shown in Fig. 5A, the major contributors to the binding free energy of OTV for WT NA were E119, D151, R292, and R371. Table 4 shows that the main component of the contribution from these residues to the binding free energy of OTV for WT NA was the electrostatic interaction. The residues S246 and E276 were in close contact with OTV at occupancies of around 50%, and no specific type of attractive interaction was identified for OTV at these residues (Table 2). The main interaction between Y406 and OTV is also close contact (35%), but additionally form a small amount of vdW interaction (9%) (Table 2). Figure 5A shows that these three closely contacting residues, S246, E276, and Y406 contributed little to the binding free energy of OTV for WT NA.

In the adjacent site of residue 274 shown in Fig. 3A, the aromatic residues, Y252 and H296 formed $\pi-\pi$ stacking interactions, and W295 formed vdW interaction with H274.

**Table 3  Occupancies at which some interactions are formed between residues in the complex of H274Y mutant influenza neuraminidase and oseltamivir (OTV).** Residue interaction is classified as hydrogen bond (HB), van der Waals interaction (vdW), stacking interaction ($\pi\pi$), and close contact (CC).

| Residue pair | HB [%] | vdW [%] | $\pi\pi$ [%] | CC [%] | Total [%] |
|---|---|---|---|---|---|
| S246-Y274 | 3 | 6 | 0 | 77 | 86 |
| Y252-Y274 | 0 | 63 | 100 | 0 | 100 |
| E276-Y274 | 99 | 0 | 0 | 0 | 99 |
| R292-Y274 | 0 | 0 | 0 | 37 | 37 |
| D293-Y274 | 0 | 24 | 0 | 60 | 84 |
| N294-Y274 | 57 | 98 | 0 | 1 | 99 |
| W295-Y274 | 0 | 76 | 50 | 5 | 84 |
| H296-Y274 | 0 | 0 | 0 | 0 | 0 |
| P301-Y274 | 0 | 5 | 0 | 88 | 93 |
| Y316-Y274 | 64 | 0 | 0 | 9 | 73 |
| E119-OTV | 81 | 7 | 0 | 2 | 83 |
| D151-OTV | 0 | 0 | 0 | 0 | 0 |
| R152-OTV | 6 | 0 | 0 | 7 | 13 |
| S246-OTV | 0 | 0 | 0 | 37 | 37 |
| E276-OTV | 0 | 0 | 0 | 48 | 48 |
| R292-OTV | 100 | 0 | 0 | 0 | 100 |
| R371-OTV | 100 | 0 | 0 | 0 | 100 |
| Y406-OTV | 1 | 18 | 0 | 42 | 62 |

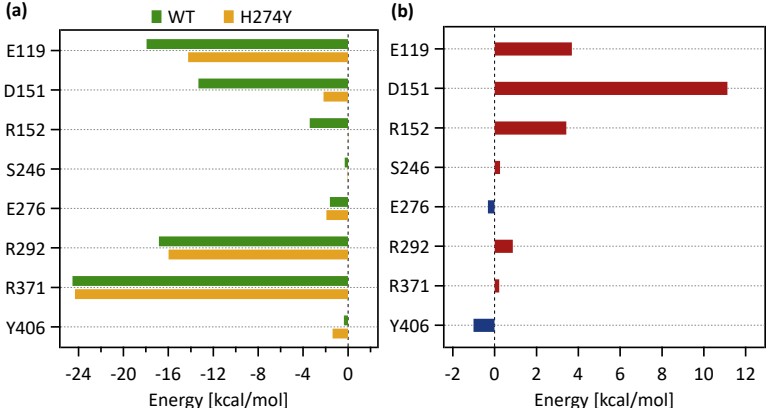

**Figure 5  Decomposed binding free energies.** (A) Decomposed binding free energy of oseltamivir (OTV) to the wild-type (WT) and H274 mutant influenza neuraminidase on the ligand binding residues obtained from the MM-PBSA calculations. (B) Changes in the decomposed binding free energy due to H274Y mutation.

Although the edges corresponding to these residue-residue pairs in Fig. 3A only represented major interactions, Table 2 shows that the contributions of both the vdW and $\pi-\pi$ stacking interactions were significant among these aromatic residues. The $\pi-\pi$ stacking interaction

**Table 4  Per-residue decomposed binding free energy of oseltamivir (OTV) to the wild-type influenza neuraminidase.** Binding free energy is decomposed into contributions from amino acid residues around OTV, which consists of van der Waals (vdW), electrostatic (ES), polar solvation, and non-polar solvation components.

| Residue | vdW (kcal mol⁻¹) | ES (kcal mol⁻¹) | Polar (kcal mol⁻¹) | Non-polar (kcal mol⁻¹) | Total (kcal mol⁻¹) |
|---------|------------------|------------------|---------------------|-------------------------|---------------------|
| E119 | 0.04 | −23.90 | 5.92 | 0.00 | −17.93 |
| D151 | −0.63 | −16.37 | 3.69 | 0.00 | −13.31 |
| R152 | −0.98 | −3.25 | 0.81 | 0.00 | −3.41 |
| S246 | −0.78 | 0.30 | 0.19 | 0.00 | −0.29 |
| E276 | −0.98 | −2.05 | 1.43 | 0.00 | −1.61 |
| R292 | −0.60 | −19.49 | 3.25 | 0.00 | −16.83 |
| R371 | 0.87 | −30.45 | 5.06 | 0.00 | −24.53 |
| Y406 | −1.40 | 1.41 | −0.37 | 0.00 | −0.37 |

**Table 5  Per-residue decomposed binding free energy of oseltamivir (OTV) to the H274Y mutant influenza neuraminidase.** Binding free energy is decomposed into contributions from amino acid residues around OTV, which consists of van der Waals (vdW), electrostatic (ES), polar solvation, and non-polar solvation components.

| Residue | vdW (kcal mol⁻¹) | ES (kcal mol⁻¹) | Polar (kcal mol⁻¹) | Non-polar (kcal mol⁻¹) | Total (kcal mol⁻¹) |
|---------|------------------|------------------|---------------------|-------------------------|---------------------|
| E119 | −0.13 | −19.47 | 5.37 | 0.00 | −14.24 |
| D151 | −0.29 | −2.66 | 0.77 | 0.00 | −2.18 |
| R152 | −0.63 | 0.30 | 0.35 | 0.00 | 0.02 |
| S246 | −0.59 | 0.43 | 0.12 | 0.00 | −0.04 |
| E276 | −0.73 | −3.12 | 1.93 | 0.00 | −1.92 |
| R292 | −0.59 | −18.61 | 3.24 | 0.00 | −15.96 |
| R371 | 0.89 | −30.53 | 5.34 | 0.00 | −24.30 |
| Y406 | −1.38 | −0.03 | 0.04 | 0.00 | −1.37 |

between Y252 and H274 was very robust, accounting for 100% of the interactions formed, whereas the formation of vdW interactions was also observed with an occupancy of about 50% (Table 2). The formation of both vdW and π−π stacking interactions was observed between H296 and H274, but the occupancy of π−π stacking interaction was 7% larger than that of vdW interaction (Table 2). The vdW interaction was mainly observed between W295 and H274 with an occupancy of about 90%, but the π−π stacking interaction was also formed between them with an occupancy of about 50%, in addition to a small amount of hydrogen-bond interaction (12%) (Table 2). The residues N294 and Y316 formed hydrogen-bond interactions with H274 (Fig. 3A). The hydrogen-bond interaction between N294 and H274 was very robust, with nearly 100% of the interaction formed, and the vdW interactions were also frequently observed between them with an occupancy of 85% (Table 2). The occupancy of hydrogen-bond interaction between Y316 and H274 was about 50% (Table 2). The residue P301 was mainly in close contact with H274 (72%), and also formed a weak vdW interaction (17%) (Table 2). The formation of both vdW interaction and close contact were observed between D293 and H274, but the occupancy of vdW interaction was

about 10% larger than that of close contact (Table 2). The residue S246 formed interactions with H274, but in much smaller fractions (Table 2).

As shown in Figure 3A, the dRIN graph indicates that residue 274 was not observed to interact directly with OTV. Therefore, the decrease in binding affinity between OTV and NA due to the H274Y mutation can be the result of indirect effects caused by changes in the residue interaction network.

## dRIN of H274Y mutant NA

As shown in Fig. 3B, the characteristics of dRIN in the H274Y mutant NA changed significantly as compared to the dRIN in the WT NA. Figure 4B indicates that the occupancy of ligand-residue interactions between the OTV and its surrounding residues, E119, D151, R152, and S246 decreased after H274Y mutation, whereas the occupancy of the OTV-Y406 interaction increased. In particular, it was observed that the occupancies of interactions with OTV were greatly reduced by more than 80% for the residues D151 and R152, and could be expected to contribute significantly to the decreased binding affinity of OTV with H274Y mutant NA. As shown in Fig. 5B, the H274Y mutation resulted in significantly increased components of decomposed binding free energy at E119, D151, and R152, destabilizing the interaction between OTV and H274Y mutant NA. D151 and R152 are residues located in the 150-loop region in NA. Previous studies have reported that the H274Y mutation in NA can change the structure of the 150-loop region from a closed to an open form (Kar & Knecht, 2012). Thus, the H274Y mutation reduced the binding affinity of OTV to NA by cleaving the ligand-residue interaction.

The dRIN structure of the H274Y mutation site remained almost unchanged, even after the His-to-Tyr mutation, except for a few residue-residue pairs and the interface bordering the OTV (Fig. 3B). In particular, two histidine moieties overlapped with each other through vdW and $\pi-\pi$ stacking interactions in the pair of H296 and H274 in WT NA, whereas no interaction was observed between the two aromatic rings in H274Y mutant NA (Fig. 3 and Tables 2 and 3). Notably, high proportions of both hydrogen-bond and vdW interactions were observed for the pair of N294 and H274 in WT NA, as shown in Table 2. However, the H274Y mutation resulted in decreased hydrogen-bond interaction occupancy compared to that in the WT NA, thereby making vdW interaction more prevalent (Tables 2 and 3). Similarly, for the interaction between D293 and residue 274, the overall occupancy of the interactions was about 80% in both the WT and H274Y mutant NA, but the preferred interaction in WT NA was the vdW interaction, and that in the H274Y mutant NA was close contact (Tables 2 and 3).

As shown in Fig. 3B, there were three residues, S246, E276, and R292, at the interface between the OTV binding site of NA and its H274Y mutation site, that were observed to interact directly with both OTV and residue 274 in the H274Y mutant NA. His-to-Tyr mutation of residue 274 caused significant changes in the dRIN at the interface region. As shown in Fig. 4B, the occupancies of interactions between residue 274 and the residues S246, E276, and R292, which form the interface, increased after the H274Y mutation, whereas the occupancies of interactions with OTV remained almost unchanged. Notably, no interaction was observed between E276 and H274 in WT NA; however, in the H274Y

mutant NA, a pair of residues E276 and Y274 formed a solid hydrogen-bond interaction with an occupancy of about 100% (Fig. 3 and Tables 2 and 3). On the contrary, there were close contacts between E276 and OTV, with an occupancy of approximately 50% in WT NA, and this occupancy remained unchanged after the H274Y mutation (Fig. 3 and Tables 2 and 3). In the pair of residues 246 and 274, weak interactions were observed between S246 and H274, with an overall occupancy of approximately 23% in WT NA, and H274Y mutation increased close contacts to approximately 80% and an overall occupancy to approximately 90% (Tables 2 and 3). In contrast, the occupancy of close contacts between S246 and OTV was 54% in WT NA; this occupancy decreased after H274Y mutation, but the difference was small, about 17% (Tables 2 and 3). The pair of residues 292 and 274 in WT NA had no interaction between R292 and H274, but that in H274Y mutant NA had close contact with an occupancy of about 40% (Tables 2 and 3). On the contrary, R292 formed a robust hydrogen-bond interaction with OTV with an occupancy of 100% in the WT NA, and this occupancy remained unchanged in the H274Y mutant NA (Tables 2 and 3).

## DISCUSSION

The results obtained from the dRIN analysis of the WT and H274Y mutant NA clearly showed that the residue interactions at both the OTV binding site and the H274Y mutation site were partially altered after the mutation. Such changes in residue interactions can reduce the binding affinity of OTV to NA, leading to drug resistance in influenza viruses. The detailed molecular mechanism of OTV drug resistance associated with the H274Y mutation in NA can be explained as follows based on the dRINs shown in Fig. 3.

There is only a limited free volume space available around residue 274, as it is surrounded by bulky aromatic residues, Y252, W295, H296, and Y316. When His is mutated to the bulkier Tyr at residue 274, the hydroxyl and phenyl groups of Y274, which is the main contributor to the bulkiness, cannot be accommodated in such a limited free volume space. The occupancies of the interactions between Y274 and its surrounding residues, including these aromatic residues, remained almost unchanged after the H274Y mutation, except for S246, E276, R292, and H296. The hydroxyl group of Y274 facing the OTV binding site forms a strong hydrogen-bond interaction with E276, one of the interface residues. Further, the phenyl group of Y274 has more close contacts with two of the interface residues, S246 and R292. However, these interface residues remain in close contact with OTV even after H274Y mutation. Overall, these interface residues slightly disrupt the orientation of OTV. As the orientation of OTV changes, the loop 150 region is pushed out of the binding region through D151 and R152. This leads to the opening of loop 150 and the loss of interaction of loop 150 residues, D151 and R152, with OTV. As shown in Fig. 5, D151 significantly contributes in stabilizing the binding between OTV and NA. Thus, it leads to a decrease in the binding free energy of the H274Y mutant NA-OTV complex. Hence, the NA-H274Y mutant becomes highly resistant to OTV.

It has been indicated that the H274Y mutation in NA causes a large shift in the position of E276 compared to WT NA, facilitating a large shift in the orientation of OTV (Collins

**(a) WT**  **(b) H274Y**

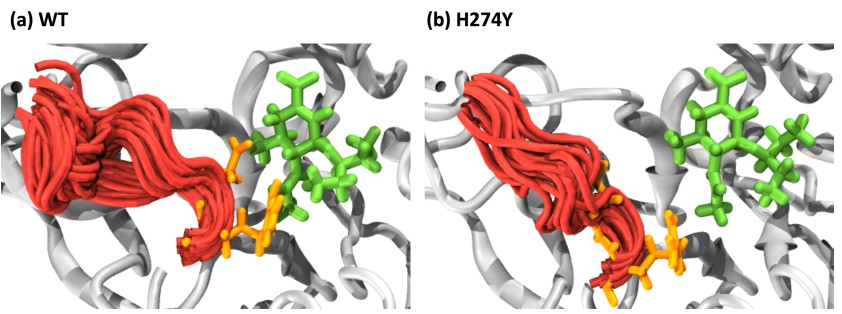

**Figure 6** **Snapshot images of the 150-loop region.** Snapshot images obtained from the MD simulations for (A) the wild-type (WT) and (B) H274Y mutant influenza neuraminidase, showing the 150-loop region (residues 147–152) and oseltamivir (OTV). The 150-loop region of NA is represented in red, and conformational changes are shown by superimposing with 100 snapshot images obtained from the MD simulation. OTV is represented in green, and the residues D151 and R152 are indicated in orange.

*et al., 2008*; *Collins et al., 2009*). As already mentioned, the 150 loop of NA is reported to open after the H274Y mutation (*Kar & Knecht, 2012*). Kar and Knecht showed that in the case of N8 NA, water molecules formed hydrogen bond bridges between OTV and binding residues in both the open and closed conformations of the 150 loop. They also showed, however, that in the case of H5N1 NA, no water-mediated binding of OTV was observed. In this study, we studied the drug resistance mechanism of the H5N1-H274Y mutant NA, and similar to their study, no water-mediated binding of OTV was observed.

We further examined how the dynamics of the 150 loop affects the drug binding mechanism of NA. Figure 6 shows the conformational changes of the 150-loop region (residues 147–152) in the WT and H274Y mutant NA by superimposing 100 snapshot images obtained from the MD simulations. As shown in Fig. 6, in the case of the WT NA, the 150-loop region tended to form closed conformations; thus, D151 and R152 frequently formed hydrogen bonds with OTV. In the case of the H274Y mutant NA, however, the 150-loop region is open, so D151 and R152 cannot form hydrogen bonds with OTV. Figure 7 shows the structural fluctuations of each residue as B-factor values for the WT and H274Y mutant NA. As shown in Fig. 7, the structural fluctuation of the 150-loop region was significantly higher in the WT NA. After the H274Y mutation, however, the structural fluctuation of the 150-loop became much smaller. In summary, Figs. 6 and 7 show that in the WT NA, the 150-loop region undergoes conformational changes with large fluctuations between the open and closed states, whereas in the H274Y mutant NA, the 150-loop remains open. This behavior of the 150-loop region is consistent with that observed in the previous study (*Kar & Knecht, 2012*), and by combining it with the dRIN analysis, the dynamic nature of the 150-loop region was quantitatively clarified in this study.

The above discussion based on the dRIN analysis supports these previous studies and provides a new perspective for comprehensively understanding the OTV drug resistance acquisition in influenza viruses associated with the H274Y mutation in NA.

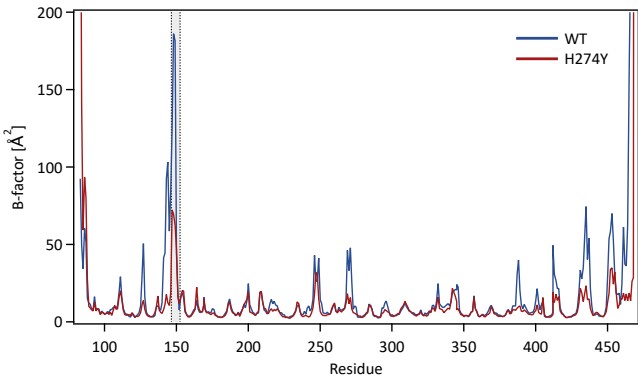

**Figure 7  B-factor values for each residue.** The structural fluctuations of each residue in the WT and H274Y mutant NA are shown as the B-factor values. The 150-loop region (residues 147–152) is represented by the gray band.

## CONCLUSIONS

In this study, we theoretically investigated the molecular mechanism of drug resistance to OTV in influenza virus with H274Y mutation of N1 NA using MD simulations. The dRIN graphs quantitatively showed how the His-to-Tyr mutation at residue 274 altered the residue interactions within the OTV binding site of NA and its H274Y mutation site. The results of dRIN analysis revealed that the OTV binding site of NA and its H274Y mutation site interact via three interface residues connecting the two sites. H274Y mutation significantly enhanced the interaction between residue 274 and the three interface residues in NA, resulting in significantly decreased interaction between OTV and its surrounding loop 150 residues. Such changes in residue interactions can reduce the binding affinity of OTV to NA, leading to OTV drug resistance in influenza viruses. In conclusion, using dRIN analysis, we succeeded in understanding the characteristic changes in residue interactions associated with the H274Y mutation, which can elucidate the molecular mechanism of OTV resistance in influenza viruses. Finally, the dRIN analysis used in this study can be widely applied to various systems such as individual proteins, protein-ligand complexes, and protein-protein complexes, to characterize the dynamic aspects of the interactions.

## ACKNOWLEDGEMENTS

We thank Prof. Silvio Tosatto for providing us with a binary version of the RING software.

### Funding

This work has received financial support from the Joint Usage/Research Center Program at Research Center for Zoonosis Control, Hokkaido University, Sapporo, Japan. The molecular simulations were carried out on the TSUBAME supercomputer, Tokyo Institute of Technology, Tokyo, Japan, under the TSUBAME Encouragement Program

for Young/Female Users. Mohini Yadav received support for living expenses and tuition fees from the Watanuki International Scholarship Foundation. The funders had no role in study design, data collection and analysis, decision to publish, or preparation of the manuscript.

### Grant Disclosures

The following grant information was disclosed by the authors:
Joint Usage/Research Center Program at Research Center for Zoonosis Control, Hokkaido University, Sapporo, Japan.
TSUBAME Encouragement Program for Young/Female Users, Tokyo Institute of Technology, Tokyo, Japan.
Watanuki International Scholarship Foundation.

### Competing Interests

The authors declare there are no competing interests.

### Author Contributions

- Mohini Yadav performed the experiments, analyzed the data, prepared figures and/or tables, and approved the final draft.
- Manabu Igarashi and Norifumi Yamamoto conceived and designed the experiments, authored or reviewed drafts of the paper, and approved the final draft.

### Data Availability

The raw data of dRIN graphs for the WT and H274Y mutant NA-OTV complexes are available in the Supplementary File, which can be viewed using Cytoscape (https://cytoscape.org).

### Supplemental Information

Supplemental information for this article can be found online at http://dx.doi.org/10.7717/peerj.11552#supplemental-information.

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
