# Peer review of "Dynamic residue interaction network analysis of the oseltamivir binding site of N1 neuraminidase and its H274Y mutation site conferring drug resistance in influenza A virus"

_PeerJ, doi:10.7717/peerj.11552_

## Round 0.1 · original submission · Major Revisions

Please address critiques of the reviewers, paying special attention to the comments of reviewer #2.

Reviewer 1 ·

Basic reporting

The article is well written.

Experimental design

I have issues with the originality of this work. This is essentially an incremental piece of work. It does not provide a substantial advance on the area described and neither does it have conceptual novelty

Validity of the findings

I have issues with the originality of this work. This is essentially an incremental piece of work. It does not provide a substantial advance on the area described and neither does it have conceptual novelty

Additional comments

The authors are recommended to have a thorough relook at this manuscript and significantly improve.

Reviewer 2 ·

Basic reporting

No comments

Experimental design

No comments

Validity of the findings

1. The 80 ns simulation time used in this study is too short of ensuring convergence. The authors have not provided the time evolution of RMSD also. Longer simulations and/or multiple copy simulations will be a better choice.
2. How the 150-loop dynamics affect the drug binding mechanisms of NA should be discussed clearly.
3. It would be helpful to show how the pocket cavity volume affects the OTV binding.
4. The disulfide bonds are essential for NA structure, and they must be careful about that, which is not clear from the methods section.
5. The binding free energies calculated with MMPBSA are shown to be very sensitive to many factors, such as the value of the dielectric constant chosen for the protein or the conformational entropy (which can be improved by running multiple replicates). The values calculated here must be interpreted very carefully.
6. The MMPBSA calculation provides the other energetic components (like vdW, Elect, Pol-sol, and non-pon-sol) in Table 1.
7. The role of molecular water in OTV recognition is missing in this text, as they play an important role in inhibitor binding.
8. A representative structure that summarizes the analysis in sections 2.1 and 2.2 would be beneficial.

Additional comments

No comment

---

## Round 0.2 · accepted · Accept

The reviewer is completely satisfied by your revision, and the amended manuscript is acceptable now..

Reviewer 2 ·

Basic reporting

This is a well-written manuscript. The conclusion is backed by simulation data and analyses.

Experimental design

The method section has been revised by the authors following the suggestion. The result can be reproduced following the protocol.

Validity of the findings

No comments

Additional comments

The authors have addressed all my queries and modified the manuscript accordingly. However, I still believe that a multiple copy simulation would be beneficial for dynamic study.